# Antireflective and Superhydrophilic Structure on Graphite Written by Femtosecond Laser

**DOI:** 10.3390/mi12030236

**Published:** 2021-02-26

**Authors:** Rui Lou, Guangying Li, Xu Wang, Wenfu Zhang, Yishan Wang, Guodong Zhang, Jiang Wang, Guanghua Cheng

**Affiliations:** 1State Key Laboratory of Transient Optics and Photonics, Xi’an Institute of Optics and Precision Mechanics of CAS, Xi’an 710119, China; lourui0423@163.com (R.L.); li_guang_ying@163.com (G.L.); wangxu@opt.cn (X.W.); wfuzhang@opt.ac.cn (W.Z.); yshwang@opt.ac.cn (Y.W.); 2School of Future Technology, University of Chinese Academy of Sciences, Beijing 100049, China; 3Electronic Information College, Northwestern Polytechnical University, Xi’an 710072, China; guodongzhang@nwpu.edu.cn (G.Z.); wjiang@nwpu.edu.cn (J.W.)

**Keywords:** femtosecond laser, antireflection, superhydrophilicity, GO/graphene

## Abstract

Antireflection and superhydrophilicity performance are desirable for improving the properties of electronic devices. Here, we experimentally provide a strategy of femtosecond laser preparation to create micro-nanostructures on the graphite surface in an air environment. The modified graphite surface is covered with abundant micro-nano structures, and its average reflectance is measured to be 2.7% in the ultraviolet, visible and near-infrared regions (250 to 2250 nm). The wettability transformation of the surface from hydrophilicity to superhydrophilicity is realized. Besides, graphene oxide (GO) and graphene are proved to be formed on the sample surface. This micro-nanostructuring method, which demonstrates features of high efficiency, high controllability, and hazardous substances zero discharge, exhibits the application for functional surface.

## 1. Introduction

Manipulating the capture and efficient use of light is extremely complex but crucially important in areas ranging from aerospace vehicles to consumer electronics [1]. This manipulating commonly involves the development of a functional surface that reduces the reflection of light and improves the absorption of energy [2,3,4]. Furthermore, surface wettability directly affects many properties, such as anti-fogging, adhesion, and self-cleaning, which, therefore, can significantly affect the performance to improve the sensitivity of devices [5,6]. The application of a hydrophilic surface can play a vital role in antifogging. This results from the property of the hydrophilic surface to ameliorate the wetting condition of the device surface by reducing the contact angles with water [7]. Hydrophilic surfaces are normally explored as anti-fogging surfaces profiting by their potential ability to rapidly spread condensing water droplets into a uniform, non-light-scattering film of water [8].

Such a dual-functional antireflective surface with a superhydrophilic effect has been found to be crucial to enhance the performance of optical, optoelectronic, and electron optical devices [9,10,11]. Surface functionality strongly depends on the material chemical composition, ambient medium, and surface topography and roughness [12]. For the surface of a particular material, surface structuring is generally more convenient for regulating the surface properties than other methods. Surface micro-nanostructuring technology, which could regulate and control the surface morphology of material, is an indispensable technology for material property modification. Therefore, such technology has drawn considerable attention and has been applied in plasmonics, optics, biochemistry, and hydrodynamics. Traditional surface microstructuring methods, including self-assembly [13], chemical vapor deposition [14,15], anode oxidation [16], soft etching [17], and photolithography [18], usually have disadvantages in efficiency and process complexity. Comparing with the above methods, femtosecond laser surface micro-nanostructuring provides an efficient, concise, and eco-friendly approach for surface modification. Typically, an intense femtosecond laser pulse is focused on a solid surface; nonlinear absorption, expansion, phase transformation and plasma evolution would subsequently occur in the focal region resulting in permanent structural or compositional changes [19,20,21,22,23]. Therefore, it is easy to induce micro/nano structures on almost all kinds of solid surfaces for femtosecond laser micro-nanostructuring.

Graphite is composed of basal plane and prismatic plane; prismatic possesses activity higher than basal. The abundant graphite materials in nature are widely used as ideal optoelectronics materials because of their excellent high strength, high modulus values, high thermal stability, low coefficient of thermal expansion, and high aspect ratio [24,25]. Since graphene was first isolated from graphite, interest in this material was increased significantly due to the coexistence of its high mobility, thermal conductivity, mechanical strength, and tunable optical properties [26]. In this aspect, graphene base surface has promising applications in surface light manipulation and surface wettability [25,27,28,29,30]. Graphene has played an irreplaceable role in the development of new gas sensors [31], modulators [32], light-emitting diodes [33], terahertz to mid-infrared photodetectors [34], and solar cells [35]. Constructing antireflective surface, improving the absorption or transmission of the incident light in a broad wavelength range are effective solutions for improving substantial sensitivity of the luminescent bolometer [36], then it can be generalized to enhance the efficiency of optoelectronic devices [37]. Femtosecond laser fabrication is diffusely used to build micro-nanostructures on account of its characteristic of quasi-nonthermal processing which depends on a relatively short pulse duration (<10^−12^ s). Preparing graphene by femtosecond laser exfoliation of graphite provides an option to produce and pattern graphene films. Nevertheless, there are still rare reports available devoted to researching the preparation of a dual-functional surface by constructing micro-nanostructures and generating graphene on graphite surface.

In this work, the graphite surface is prepared by femtosecond laser pulses for the fabrication of dual-functional antireflective surface with a superhydrophilic effect which could achieve both the efficient absorption of light and complete spread of droplets. Micro-nanostructures are fabricated on graphite surfaces via the femtosecond laser ablation. The morphology and formation mechanism of the structures are carefully studied. Through varying the scanning intervals and the pulse energy, low broadband reflectance and superhydrophilic effects are well realized. Furthermore, the process of generating GO/graphene is explored in detail by analyzing the Raman spectra and x-ray photoelectron spectroscopy (XPS) of the surfaces induced by different pulse energies.

## 2. Experimental

### 2.1. Materials

In our experiment, graphite foil was divided into square flakes (25 mm × 25 mm × 0.1 mm in dimension) for easier processing. Prior to laser treatment, the upper surface of the graphite foil was cleaned with ethanol.

### 2.2. Fabrication of Functional Surface

A one-step fabrication strategy was prepared to induce the micro-nanostructure on the graphite foil sample. Figure 1 shows the schematic diagram and operation mechanism of the femtosecond laser fabrication. The Yb: KGW femtosecond oscillator-amplifier system (Pharos, Light Conversion, Vilnius, Lithuania) is capable of generating a central 1030 nm wavelength, 220 fs pulse duration, 200 uJ maximum energy pulse laser (0.26 to 2.58 uJ used in this experiment) and a repetition rate of 100 kHz. The laser energy was regulated by a combination of a polarization splitting prism and a half-wave plate. A microscope objective (5×, NA = 0.14, f = 40 mm, Mitutoyo objective, Kanagawa, Japan) and a three-dimensional mobile platform (ANT, Aerotech, Pittsburgh, PA, USA) was applied to focus the laser beam to form a 3.5 μm spot size and move the graphite samples with 15 mm/s scanning velocity along a preset route to complete the process.

### 2.3. Characterization and Measurement

Field emission scanning electron microscopy (Verios G4, FEI, Hillsboro, OR, USA) was utilized to characterize the laser-processed graphite covering with micro-nanostructures. A spectrophotometer system incorporated with an integrating sphere of 150 mm in diameter (PerkinElmer Lambda1050, PerkinElmer, Inc., Waltham, MA, USA) was used to characterize the wavelength dependence of the hemispherical reflectance in the ultraviolet, visible and near infrared regions (250 to 2250 nm). The fluctuations of the measured spectral curves around 860 nm are due to the switching of the detectors at this band. The static contact angle of the deionized water droplet (1 μL) on sample surface was measured using the sessile drop method. The charge-coupled device (CCD) camera was used to capture the droplet image on the graphite surface to provide contact-angle information. The three-dimensional morphologies of the sample surfaces were characterized using a confocal microscope. The Raman excitation spectra of the samples were characterized by a microscopic confocal Raman spectrometer. The XPS was used to determine the content of different functional groups in the laser-induced structures.

## 3. Results and Discussion

### 3.1. Surface Reflectance

The focused femtosecond laser has an extremely high intensity in the range TW/cm^2^ and is universally expected to be adept at building micro or nanostructures by laser surface ablation. Comparison of the graphite surface reflectance values corresponding to different intervals are shown in Figure 2a. The reflectance spectrum of the untreated graphite is offered in black as a reference. Excellent antireflection is achieved on originally high reflective gray graphite surfaces by means of laser induction. The scanning interval between two adjacent traces has an appreciable impact on the reflectance. The optimal reflectance is obtained with a scanning interval of 25 μm, being measured with below 2% reflectivity in the ultraviolet and visible spectral regions and below 5% reflectivity on average in the ultraviolet and visible, and near infrared regions (250 to 2250 nm). The surface reflectance with a scanning interval of 25 μm not only remains a low value of reflectance in the ultraviolet and visible spectral regions, but also maintains a comparatively low value extending the measured near infrared region. Due to the extremely reduced reflectance in the visible region; i.e., below 1.2%; the original gray surface of the graphite sample translated to be pitch black, as shown in the inset of Figure 2a. Femtosecond laser fabrication with optimal parameters of scanning intervals dramatically decreases the reflectance of the graphite surfaces in a very broad spectrum from 250 nm to 2250 nm.

The morphologies of different scanning intervals prepared by multi-femtosecond laser pulses on the graphite surface are characterized as illustrated in Figure 2b–e, corresponding to intervals of 100 μm, 75 μm, 50 μm and 25 μm, respectively. The laser fluence is 2.6 J/cm^2^ (E = 0.93 μJ). Graphite ejections are observed, existing on either side of the laser scan track. The morphologies of the ejections reveal that it has experienced a melting and quick re-solidifying process during laser preparation. Corresponding to the scanning interval of 100 μm, 75 μm and 50 μm, the distribution of laser-induced structures is limited to both sides of the scanning track. Unlike with the above situation, the laser-induced structure is able to cover each machined mesh unit as the scanning interval decreases to 25 μm. Gridlines micron grooves are induced in femtosecond laser scanning tracks; meanwhile, the lattices divided by the micron grooves are also completely covered by the micro-nano composite structures.

The reflectance spectra of the graphite surfaces with femtosecond laser pulse energies ranging from 0.52 μJ to 2.06 μJ (with a fixed scanning interval of 25 μm) are shown in Figure 3. A slight increase in reflectance is observed at a non-optimal laser pulse energy. A significant decrease in reflectance and increase in absorptance over the broadband spectrum are observed after optimizing the parameters of femtosecond laser pulse energy. As the laser pulse energy gradually reaches 1.52 μJ, the average reflectance of graphite surface decreases to 2.7% in the ultraviolet, visible and near infrared regions (250 to 2250 nm). Compared with the optimization result of the laser scanning interval as previously mentioned, it is apparent that the reflectance of the graphite surface after laser energy re-optimization is obviously suppressed over the whole of the measurement wavelength. The spectral curve of the lowest reflectance is provided in orange, with below 1.2% reflectivity in the ultraviolet and visible spectral region and below 3% reflectivity on average in the ultraviolet, visible, and near infrared regions (250 to 2250 nm).

With the laser pulse energy increasing from smaller to greater, the broadband reflectance of the graphite surface decreases gradually and then increases after reaching a certain value. Figure 4 shows the SEM images of graphite surface structures created with different laser pulse energies. Figure 4a–c, respectively, correspond to the morphologies of prepared surfaces with scanning laser energies of 0.93 μJ, 1.52 μJ and 2.06 μJ, while Figure 4d–f correspond to a local magnification of Figure 4a–c. The evolution of micron grooves etched by femtosecond laser with different laser energy is observed. While the femtosecond laser etched micron-scale grooves on the scanning track to provide geometrical light trapping, and the micro-nano composite structures are induced on both sides of the grooves. It is known that when the structure size is much larger than the characteristic wavelength, the surface reflectance can be predicted using the ray tracing method in the limit of geometrical optics, which attributes the reflective inhibition to the multiple reflection effects of the surface structure. Under this premise, the reflectance decreases as the surface structure is higher (or deeper). For the multilayer ladder-shaped structure, the parallel cross distributed grooves can efficaciously assemble incident light of the broad-spectrum from all direction. Thereafter, the incident light is multiply reflected in the layered grooves to enhance light absorption.

Except for constructing the micron-scale grooves, femtosecond laser direct writing induces abundant nanostructures on the graphite surface as well. The mechanism responsible for the surface wave generation is appropriate to be explained by the surface plasmon polaritons (SPPs) in many researches [38,39,40,41,42]. The formation of the subwavelength periodic surface structure of different materials by femtosecond laser irradiation mainly results in the interference between the incident laser and the polarization wave of the surface plasma excited by the material surface [43].

The interference between the electromagnetic field of the SPPs and the incident laser pulse results in the deposition of light energy in a spatially modulated manner in the electronic system of the material [44]. Nanostructures can be clearly observed on the surface of samples induced by different pulses of energy. Three different types of nanostructures are compared as illustrated in Figure 4d–f. The villous nanostructures are induced by 0.93 μJ of pulse energy, and the stripy nanostructures are induced by 2.06 μJ of pulse energy. Different from the above two cases, a novel type of nano-stripes covered with nano-villi is observed on the preparation surface induced by 1.52 μJ of pulse energy. As can be seen from the reflectance curve (orange) in Figure 3, the composite structure of nano-stripes and nano-villi plays a superposition role in the absorption of incident light. A reasonable explanation is that compared with the other two nanostructures, the novel composite nanoscale structures can act as effective medium layers to mitigate the optical impedance mismatch between the material surface and air environment.

### 3.2. Surface Wettability

The dependence of the static contact angles on the scanning intervals at 25 °C (room temperature) is studied as shown in Figure 5a. The untreated graphite surface is defined to have an interval of 5000 μm, which is larger than the scale of measurement droplet. It could be seen that the static contact angle firstly decreases with the decreasing of the interval in the range of 100 μm to 50 μm, and then reaches the minimum value of 0° (ie. complete wetting) about five seconds after it hits the surface at the interval of 50 μm and 25 μm. A live video has been recorded to show the droplet from falling onto the surface to complete wetting. The untreated graphite surface is characterized to be hydrophilic with a contact angle of 78°, which is consistent with other reports [45]. It reveals that the wettability of the graphite sample experienced a transformation from hydrophilicity to super-hydrophilicity in the interval-scanning process. This wettability transformation could be explained by the Wenzel model [46], as it explains that the existence of a rough surface makes the actual solid–liquid contact area larger than the apparent geometric contact area and predicts that the contact angle will decrease with increasing roughness of a hydrophilic surface. The relationship between the static contact angle and pulse energy is studied under a fixed interval of 25 μm, as shown in Figure 5b. It exhibits that the contact angle reaches 0° on the surfaces prepared by four different pulse energies; the deionized water droplets are completely spread out on the surface of the sample. Furthermore, in order to explore the wettability of surface structures under different temperature environments, the static contact angle is studied at 80 °C and 3 °C, as shown in Figure 5c–f. It is worth paying attention to the fact that the dependence of the static contact angles on the scanning intervals and pulse energy has not changed significantly under high and low temperature environments. Similarly, complete superhydrophilicity wetting is achieved by adjusting laser parameters on surfaces with a 25 μm scanning interval. The stable superhydrophilicity provides a basis for the anti-fogging application of antireflective materials in high and low temperature environments.

### 3.3. Raman and XPS Analysis

Figure 6a shows the 3D morphology image of surface structures created with the processing parameters of scanning interval of 25 μm and pulse energy of 1.52 μJ. The surface of this processing parameter exhibits the optimal broadband antireflectivity. The entire initial surface is first processed into a ridge-like surface by parallel femtosecond laser scanning tracks, then the ridge-like surface is segmented into units covered with abundant micro-nano structures by vertically crossed laser scanning tracks. The sample surface is covered with this type of micro-nanostructure to realize a dual function. To explore the contribution of the formation of new substances on the functional properties of the sample surface, Raman spectra and XPS were introduced to detect the formation of new substances on the graphite surface.

The samples prepared by different femtosecond laser energies are characterized by Raman spectroscopy which were excited by a 532 nm laser source to gain useful information on their molecular structure. The G-band in general arises due to the in-plane vibrations of the sp^2^ carbon atoms. The D-band arises due to the disorder and/or defects in the sp^2^ bonded carbon atoms [47,48,49]. Representative Raman spectra of sample surfaces at different laser pulse energies are shown in Figure 6b together with that of the untreated graphite surface. The black curve shows the Raman spectrum of the untreated graphite, which displayed a G-band at ~1580 cm^−1^ and a broad doublet structure 2D-band at ~2700 cm^−1^. The G- and D-band vibrations of the features of GO and graphene are observed as shown in curve 2 to curve 7. Compared to the Raman spectrum of the original sample, those of the prepared-by-laser-exfoliation surface have three peaks at ~1350 cm^−1^, ~1580 cm^−1^, ~2700 cm^−1^ corresponding to the D-band, G-band, 2D-band, respectively. The curves show that upon induction the value of the disorder parameter defined by the ID/IG ratio initially increases (0.26 μJ, 0.52 μJ, and 0.93 μJ treatment of the samples) and decreases with higher energy induction (1.52 μJ, 2.06 μJ, and 2.58 μJ treatment) [50]. The D-band intensity presents a non-monotonous behavior (increases at first and then decreases), which displayed that the original graphite is oxidized to GO with an increase in the disorder and subsequently reduced to graphene with a decrease in the disorder.

The C 1s XPS spectra are fitted according to the peak position as show in Figure 7a–g. Figure 7a–g correspond to the C 1s XPS spectra of the surface structure generated by the induced energy of 0 μJ (untreated graphite surface), 0.26 μJ, 0.52 μJ, 0.93 μJ, 1.52 μJ, 2.06 μJ, and 2.58 μJ, respectively. The variation curves of the contents of the three different functional groups with the pulse energy are illustrated in Figure 7h. It could be obtained in the following regulation that the contents of three functional groups varied in different trends with the increase of the induced laser pulse energy. Accompanying the laser pulse energy increases from 0 to 0.93 μJ, the content of the C–C functional group functional groups of the sample surface calculated by quantitative analysis of XPS peaks decreases from 60.30 to 37.45%. Correspondingly, the content of the C–O functional group increases from 28.68 to 46.61% and the content of the C=O functional group mildly increases from 11.02 to 15.94%. This indicates that the original graphite surface reacts with oxygen elements in the atmosphere to convert to GO under the action of laser pulse injection.

The laser pulse energy continues to increase from 0.93 μJ to 2.58 μJ; the signal intensity of the C–C functional group with SP^2^ hybrids represents a rapid increase. The quantitative analysis of XPS peaks reveals that the content of the C–C functional group increases from 37.45 to 64.17 %, while the content of the C–O functional group functional groups decreases from 46.61 to 20.25% and the content of the C=O functional group remains basically constant. This indicates that graphite on the sample surface reacts with oxygen to form GO which is then reduced to graphene by high intensity laser pulse. The above results prove that GO and graphene could be generated on the graphite surface under laser pulse exfoliation and also verify the results acquired by the Raman spectra. Overall, the present work indicates that the dual-functional surface is due to the synergistic effect of micro-nanostructures and GO/graphene. It also demonstrates the prospective applications of the femtosecond laser induction strategy for the preparation of surface functional components.

## 4. Conclusions

In summary, we presented and experimentally verified a project of facile fabrication of a dual-functional antireflective surface with a superhydrophilic effect based on femtosecond laser writing. The graphite surface is covered with micro-nanostructures by this method. The micro-nanostructuring mechanism is attributed to the multi-pulse femtosecond laser ablation and SPPs. This micro-nanostructuring method in one step is demonstrated to have the features of high efficiency, high controllability, and eco-friendliness. The dependence of broadband reflectance on the scanning interval and pulse energy are carefully studied. The reflection of the surface covered by micro-nanostructures is identified to be less than 2.7% in the ultraviolet, visible and near-infrared regions. The superhydrophilic effect is excellently achieved on the low-broadband reflectance surface in three different temperature environments simultaneously. The Raman spectra and XPS explain that a new carbon form, GO/graphene, is generated on the dual-functional surfaces.

## Figures and Tables

**Figure 1 micromachines-12-00236-f001:**
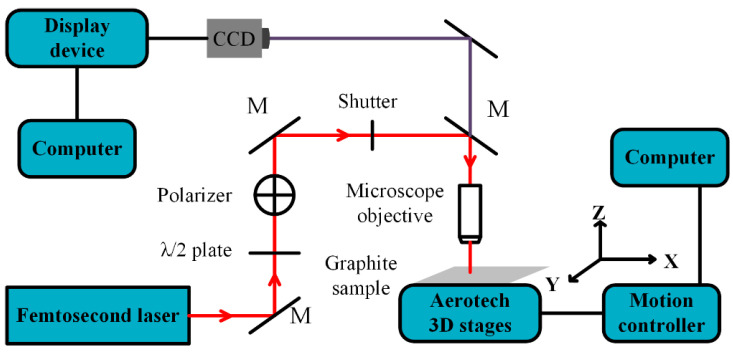
Schematic of the femtosecond laser processing setup for functional surface.

**Figure 2 micromachines-12-00236-f002:**
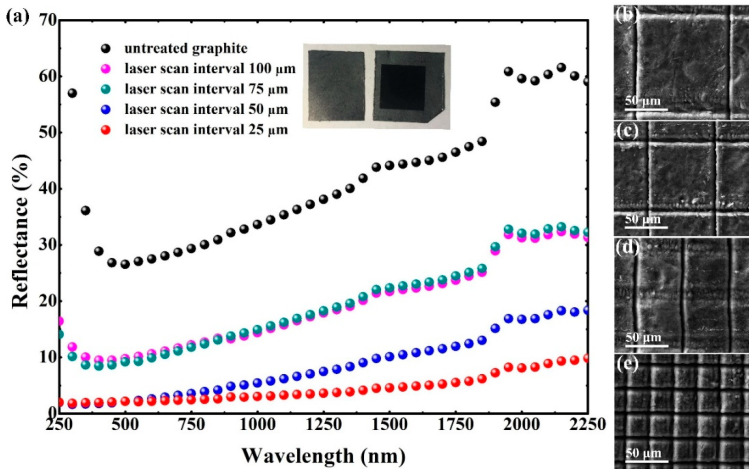
(**a**) Comparison of the graphite surface reflectance values corresponding to different intervals; scanning electron microscope (SEM) images of the laser-induced structures characterizing different scanning intervals of (**b**) 100 μm, (**c**) 75 μm, (**d**) 50 μm, and (**e**) 25 μm, respectively.

**Figure 3 micromachines-12-00236-f003:**
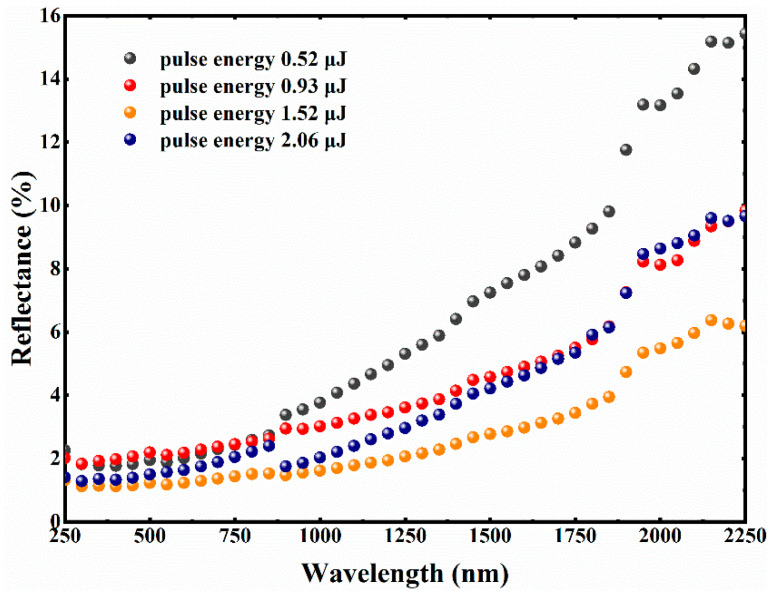
Comparison of the graphite surface reflectance from different scanning pulse energy.

**Figure 4 micromachines-12-00236-f004:**
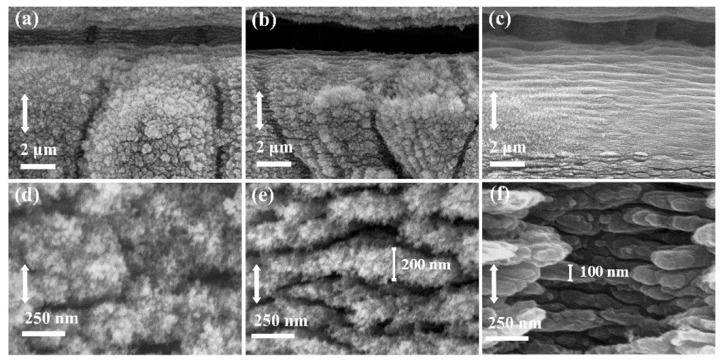
SEM images of surface structures. Here, the laser pulse energy for sample (**a**–**c**) is 0.93 μJ, 1.52 μJ and 2.06 μJ, respectively; (**d**–**f**) are high-magnification SEM image corresponding to (**a**–**c**). Double headed solid arrow represents incident laser polarization.

**Figure 5 micromachines-12-00236-f005:**
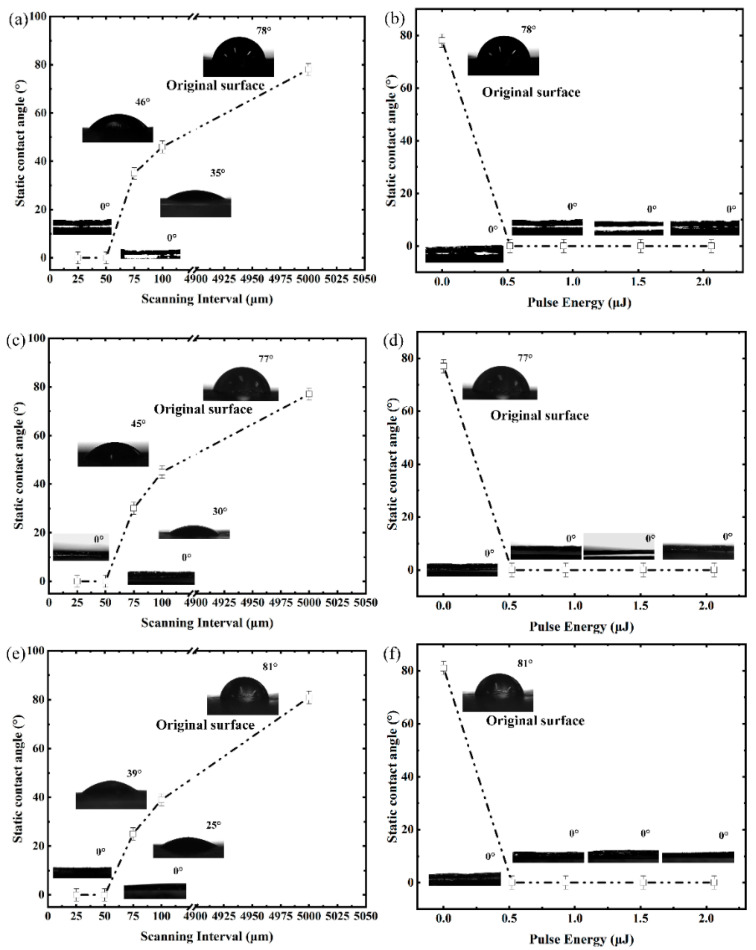
Dependence of the static contact angles on scanning interval at (**a**) 25 °C (**c**) 80 °C (**e**) 3 °C and dependence of the static contact angles on pulse energy at (**b**) 25 °C (**d**) 80 °C (**f**) 3 °C.

**Figure 6 micromachines-12-00236-f006:**
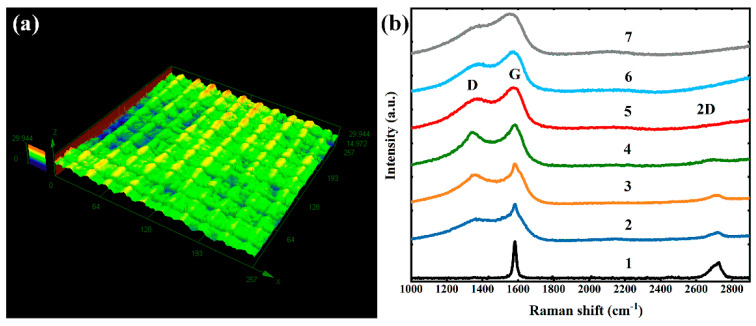
(**a**) Surface structure fabricated at a pulse energy of 1.52 μJ characterized by confocal laser scanning microscope and (**b**) Raman spectrum of the sample at different laser pulse energy: 1, untreated graphite surface; 2, 0.26 μJ; 3, 0.52 μJ; 4, 0.93 μJ; 5, 1.52 μJ; 6, 2.06 μJ; 7, 2.58 μJ.

**Figure 7 micromachines-12-00236-f007:**
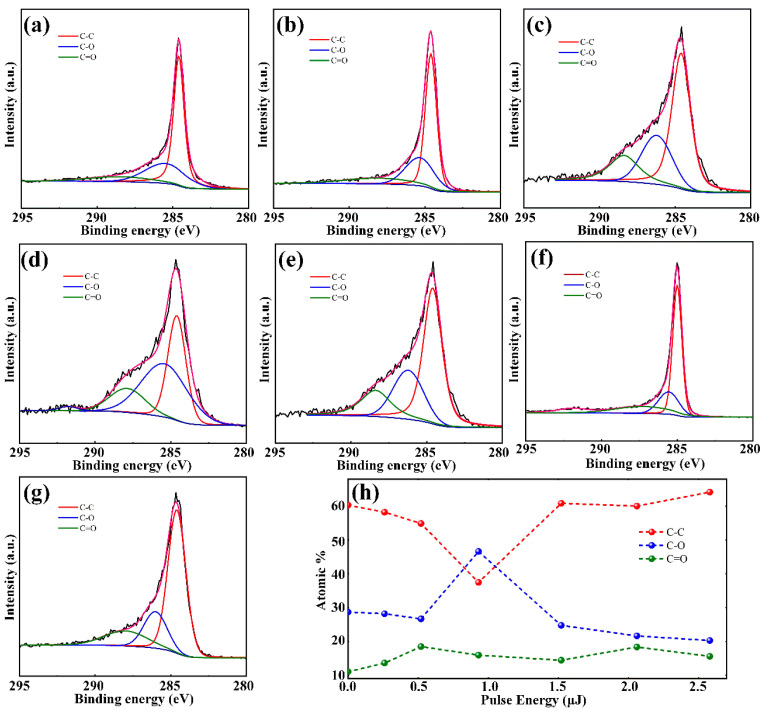
C 1s XPS spectra of the sample at different laser pulse (**a**) untreated graphite surface; (**b**) 0.26 μJ; (**c**) 0.52 μJ; (**d**) 0.93 μJ; (**e**) 1.52 μJ; (**f**) 2.06 μJ; (**g**) 2.58 μJ. (**h**) Comparison of the contents of three functional groups changing with pulse energy.

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
