# Peer review of "Antireflective and Superhydrophilic Structure on Graphite Written by Femtosecond Laser"

_micromachines, 2021, doi:10.3390/mi12030236_

Round 1

Reviewer 1 Report

The manuscript ID micromachines-1113287 titled “Antireflective and superhydrophilic structure on graphite written by femtosecond laser” presents an experimentally work concerning an antireflective and superhydrophilic graphite surface induced by means of femtosecond laser writing.

The aim of this work is described clearly; the experiments are well carried out and detailed shown in the manuscript. After minor revisions, the manuscript will be ready for publication. Below are reported my comments and suggestions to improve the reading of the manuscript adding more value to the scientific community and Micromachines readers making it suitable for publication:

- In the fabrication of functional surface section, the Authors should add more detail: the numerical aperture of the microscope objective, the scanning speed, the laser polarization on the target with respect to the scan direction, the laser spot waist, the used laser frequency.

- In the Results and Discussion the sentence “Femtosecond laser fabrication is diffusely used to build micro-nanostructures on account of its characteristic of quasi-nonthermal processing which depends on the relatively short pulse duration (<10-12s).” is better suited for the introduction part. Moreover, I suggest rewriting the paragraph relative to Fig. 2 because the Authors declare first the result: “Femtosecond laser fabrication with optimal parameters of scanning intervals dramatically decreases the reflectance of the graphite surfaces in a very broadband spectrum from 250 nm to 2250 nm as illustrated in Figure 2a.” and then the fabrication process with laser fluence and scanning interval (between two adjacent traces) used. Correct the superscript in 2.6 J/cm2. Anyway, correct the superscripts and the subscript in the full manuscript.

- The Authors should indicate the fixed scanning interval used for the experiments of Fig. 3.

- As regards the “… micro-nano composite structures are induced on both sides of the grooves.”, in particular the cited “nano-stripes”, these structures are often referred in the literature as LIPSS (laser-induced periodic surface structures) [Ref. 41] which orientation is relative to the laser polarization (P), in particular HSFL (high-spatial frequency LIPSS). For this reason, it is interesting to highlight the laser polarization state on the SEM figures. Are there differences between the LIPSS morphologies along the vertical and the horizontal scan tracks?

- The section regarding the surface plasmon polariton wavenumber KSPP reports the equations from the literature without adding further information, e.g. comparing the theoretical and the experimental spatial dimensions of the nano-stripes. Is this part fundamental for this work? (Reduce or discuss this section).

- Increase the size of the axes of Fig. 7 to improve the readability.

- In the Introduction section, the Authors should check the Ref.12 associated with the sentence: “Surface functionality strongly depends on the material chemical composition, ambient medium, and surface topography and roughness [12].” Is such reference appropriate? The authors should consider expanding the state-of-the-art regarding the structures induced by femtosecond laser which impact on the surface wettability and on the graphene functionalization: [Nano Research, 13 (2020), pp. 2332–2339, https://doi.org/10.1007/s12274-020-2852-3] and [Appl. Surf. Sci., 494 (2019), pp. 1055-1065, https://doi.org/10.1016/j.apsusc.2019.07.126]. The rest of the references are properly cited in the manuscript.

Author Response

Response to Reviewer 1 Comments

Point 1: In the fabrication of functional surface section, the Authors should add more detail: the numerical aperture of the microscope objective, the scanning speed, the laser polarization on the target with respect to the scan direction, the laser spot waist, the used laser frequency.

Response 1: We thank the reviewer for the valuable comment. We had added more detail (the numerical aperture of the microscope objective, the scanning speed, the laser polarization on the target with respect to the scan direction, the laser spot waist, the used laser frequency) in the new version.

Point 2: In the Results and Discussion the sentence “Femtosecond laser fabrication is diffusely used to build micro-nanostructures on account of its characteristic of quasi-nonthermal processing which depends on the relatively short pulse duration (<10-12s).” is better suited for the introduction part. Moreover, I suggest rewriting the paragraph relative to Fig. 2 because the Authors declare first the result: “Femtosecond laser fabrication with optimal parameters of scanning intervals dramatically decreases the reflectance of the graphite surfaces in a very broadband spectrum from 250 nm to 2250 nm as illustrated in Figure 2a.” and then the fabrication process with laser fluence and scanning interval (between two adjacent traces) used. Correct the superscript in 2.6 J/cm2. Anyway, correct the superscripts and the subscript in the full manuscript.

Response 2: We thank the reviewer for the valuable comment. We had adjusted the sentence “Femtosecond laser fabrication is diffusely used to build micro-nanostructures on account of its characteristic of quasi-nonthermal processing which depends on the relatively short pulse duration (<10-12s).” to the introduction part. We had adjusted and rewritten the paragraph relative to Fig. 2 as below.

Comparison of the graphite surface reflectance values corresponding to different in-tervals are shown in Figure 2a. The reflectance spectrum of the untreated graphite is offered in black as a reference. Excellent antireflection is achieved on originally high reflective gray graphite surfaces by means of laser induction. The scanning interval between two adjacent traces has appreciable impact on the reflectance. The optimal reflectance is obtained with a scanning interval of 25 μm, being measured with below 2% reflectivity in the ultraviolet and visible spectral regions and below 5% reflectivity on average in the ultraviolet and visible, and near infrared regions (250-2250 nm). The surface reflectance with a scanning interval of 25 μm not only remains a low value of reflectance in the ultraviolet and visible spectral regions, but also maintains on a comparatively low value extending the measured near infrared region. Due to the ex-tremely reduced reflectance in the visible region, i.e., below 1.2%, the original gray surface of the graphite sample translated to be pitch black, as shown in the inset of Figure 2a. Femtosecond laser fabrication with optimal parameters of scanning inter-vals dramatically decreases the reflectance of the graphite surfaces in a very broad-band spectrum from 250 nm to 2250 nm.

The superscript in 2.6 J/cm2 and the superscripts and the subscript in the full manuscript had been corrected in the new version.

Point 3: The Authors should indicate the fixed scanning interval used for the experiments of Fig. 3.

Response 3: We thank the reviewer for the valuable comment. We had indicated the fixed scanning interval used for the experiments of Fig. 3. (page 4 in the new version)

Point 4: As regards the “… micro-nano composite structures are induced on both sides of the grooves.”, in particular the cited “nano-stripes”, these structures are often referred in the literature as LIPSS (laser-induced periodic surface structures) [Ref. 41] which orientation is relative to the laser polarization (P), in particular HSFL (high-spatial frequency LIPSS). For this reason, it is interesting to highlight the laser polarization state on the SEM figures. Are there differences between the LIPSS morphologies along the vertical and the horizontal scan tracks?

Response 4: We thank the reviewer for the valuable comment. We had added the direction of the laser polarization and nanostructure period in Fig.4 in new version. The LIPSS morphologies along the vertical and the horizontal scan tracks are similar.

Point 5: The section regarding the surface plasmon polariton wavenumber KSPP reports the equations from the literature without adding further information, e.g. comparing the theoretical and the experimental spatial dimensions of the nano-stripes. Is this part fundamental for this work? (Reduce or discuss this section).

Response 5: We thank the reviewer for the valuable comment. We used the equations (1) and (2) to show the dispersion relation of surface plasmon polariton wavenumber Kspp, which could explain the formation of nanostripes. Indeed, this section did not contribute significantly to analysis surface functional results in this paper. We had reduced this section in the new version.

Point 6: Increase the size of the axes of Fig. 7 to improve the readability.

Response 6: We thank the reviewer for the valuable comment. We had increased the size of the axes of Fig. 7 to improve the readability in the new version. (page 10 in the new version)

Point 7: In the Introduction section, the Authors should check the Ref.12 associated with the sentence: “Surface functionality strongly depends on the material chemical composition, ambient medium, and surface topography and roughness [12].” Is such reference appropriate? The authors should consider expanding the state-of-the-art regarding the structures induced by femtosecond laser which impact on the surface wettability and on the graphene functionalization: [Nano Research, 13 (2020), pp. 2332–2339, https://doi.org/10.1007/s12274-020-2852-3] and [Appl. Surf. Sci., 494 (2019), pp. 1055-1065, https://doi.org/10.1016/j.apsusc.2019.07.126]. The rest of the references are properly cited in the manuscript.

Response 7: We thank the reviewer for the valuable comment. We had used new reference [12] [J. Appl. Phys. 117(2015), pp. 137–171. https://doi.org/ 10.1063/1.4905616.] to replace the previous one to associate with the sentence: “Surface functionality strongly depends on the material chemical composition, ambient medium, and surface topography and roughness.”

We had added [29] [Nano Research, 13 (2020), pp. 2332–2339, https://doi.org/10.1007/s12274-020-2852-3] and [30] [Appl. Surf. Sci., 494 (2019), pp. 1055-1065, https://doi.org/10.1016/j.apsusc.2019.07.126] to expand the state-of-the-art regarding the structures induced by femtosecond laser which impact on the surface wettability and on the graphene functionalization.

Reviewer 2 Report

This manuscript presents micro-nano structures on the graphite surface by the femtosecond laser scanning process. The results provide some useful information for the functional surface application, but there are several minor issues that must be addressed before this article meets the criteria of publication:

  1. The laser focus spot size and the NA of the microscope objective should be added.
  2. How to calculate the laser fluence 2.6 J/cm2 from pulse energy 0.93 μJ? Is the laser beam type is Gaussian?
  3. P4, Line 144 “the distribution of laser-induced structures is limited to both sides of the scanning track.”. The magnified image should be added to explain what laser-induced structures is? Besides, the direction of the electric field and nanostructure period should be added. Explain the reasons for nanostructure forming, whether there are other possible reasons besides SPP.

Author Response

Response to Reviewer 2 Comments

Point 1: The laser focus spot size and the NA of the microscope objective should be added.

Response 1: We thank the reviewer for the valuable comment. We had added the laser focus spot size and the NA of the microscope objective in the new version.

Point 2: How to calculate the laser fluence 2.6 J/cm2 from pulse energy 0.93 μJ? Is the laser beam type Gaussian?

Response 2: We thank the reviewer for the valuable comment. The laser beam type is Gaussian. The femtosecond laser passing a 5× objective was directly focused on the sample surface with a radius of 3.36 μm. The theoretical beam diameter of a laser beam after passing through the focusing lens is calculated by the following equation

where f is the focal length of the objective, λ is the wavelength of the laser, W is the input/collimated laser beam diameter onto the objective, and M2 is the beam propagation factor, which describes the difference between a real laser beam and an ideal diffraction-limited Gaussian beam. By the above formula, the focus spot area can be calculated as 35.54 μm2. In the end, the laser fluence 2.6 J/cm2 can be calculated from pulse energy 0.93 μJ.

Point 3: P4, Line 144 “the distribution of laser-induced structures is limited to both sides of the scanning track.”. The magnified image should be added to explain what laser-induced structures is? Besides, the direction of the electric field and nanostructure period should be added. Explain the reasons for nanostructure forming, whether there are other possible reasons besides SPP.

Response 3: We thank the reviewer for the valuable comment. The magnified image had been shown in Fig.4 to explain laser-induced structures. The direction of the nanostripes is perpendicular to the direction of laser polarization, and the period of the nanostripes is smaller than the laser wavelength. We had added the direction of the laser polarization and nanostructure period in Fig.4 in new version. The formation mechanism of nanostructure is a complex process, SPP is one of the theoretical models. There are several other mechanisms suggested to explain the formation of LIPSS such as the generalized scattering and interference (Sipe) model, self-organization of unstable matter mediated LIPSS formation.

Reviewer 3 Report

My remarks and suggestions:

  1. Line 92 , the unit of  pulse laser energy "200uJ", should be corrected.
  2. In the text (also in caption of figures)  the authors should give more  laser beam parameters used during laser treatment of graphite samples  (laser spot diameter, laser fluency, scan velocity,  pulse repetition frequency).

  3. Authors  described all results obtained by different experimental method in  one section (3 Results and discussion). Please divide you section 3 in to subsection e.g. (optical reflectivity, wettability, Raman and XPS analysis…).

  4. In line 230 authors wrote “The untreated graphite surface is characterized to be hydrophilic with 229 a contact angle of 78°, which is consistent with other reports.” Please give proper  literature for this suggestion.

  5. What is the purpose of rewriting from literature [42 43]  the equations  (1) and (2) if they are not used for analysis yours  results in this paper?

Author Response

Response to Reviewer 3 Comments

Point 1: Line 92, the unit of pulse laser energy "200 uJ", should be corrected.

Response 1: We thank the reviewer for the valuable comments on our manuscript. The pulse laser energy "200 uJ" is the maximum energy of a sing pulse that can be provided by the laser equipment used in the experiment. In this case, the Pharos laser system provide an average power of 20 W at 100 kHz pulse repetition frequency. And we had added the laser energy range used in the experiment in the manuscript.

Point 2: In the text (also in caption of figures) the authors should give more laser beam parameters used during laser treatment of graphite samples (laser spot diameter, laser fluency, scan velocity, pulse repetition frequency).

Response 2: We thank the reviewer for the valuable comment. We had added more laser beam parameters used during laser treatment of graphite samples (laser spot diameter, laser fluency, scan velocity, pulse repetition frequency). (page 2, 3 in the new version)

Point 3: Authors described all results obtained by different experimental method in one section (3 Results and discussion). Please divide you section 3 in to subsection e.g. (optical reflectivity, wettability, Raman and XPS analysis…).

Response 3: We thank the reviewer for the valuable comment. We had divided the section 3 into three subsections: Surface Reflectance, Surface Wettability, Raman and XPS analysis. (in the new version)

Point 4: In line 230 authors wrote “The untreated graphite surface is characterized to be hydrophilic with 229 a contact angle of 78°, which is consistent with other reports.” Please give proper literature for this suggestion.

Response 4: We thank the reviewer for the valuable comment. We had added proper literature [45] (DOI 10.1016/j.surfcoat.2017.03.047) for this suggestion in the new version.

Point 5: What is the purpose of rewriting from literature [42 43] the equations (1) and (2) if they are not used for analysis yours results in this paper?

Response 5: We thank the reviewer for the valuable comment. We used the equations (1) and (2) to show the dispersion relation of surface plasmon polariton wavenumber Kspp, which could explain the formation of nanostripes. Indeed, this section did not contribute significantly to analysis surface functional results in this paper. It happened that another reviewer suggested us to reduce this part. We had reduced this section in the new version and will make a more in-depth study of this content in the future work. 
